# Complex reaction processes in combustion unraveled by neural network-based molecular dynamics simulation

Jinzhe Zeng [1], Liqun Cao [1], Mingyuan Xu[1], Tong Zhu [1,2]✉ & John Z. H. Zhang [1,2,3,4]✉

Combustion is a complex chemical system which involves thousands of chemical reactions and generates hundreds of molecular species and radicals during the process. In this work, a neural network-based molecular dynamics (MD) simulation is carried out to simulate the benchmark combustion of methane. During MD simulation, detailed reaction processes leading to the creation of specific molecular species including various intermediate radicals and the products are intimately revealed and characterized. Overall, a total of 798 different chemical reactions were recorded and some new chemical reaction pathways were discovered. We believe that the present work heralds the dawn of a new era in which neural network-based reactive MD simulation can be practically applied to simulating important complex reaction systems at ab initio level, which provides atomic-level understanding of chemical reaction processes as well as discovery of new reaction pathways at an unprecedented level of detail beyond what laboratory experiments could accomplish.

[1] Shanghai Engineering Research Center of Molecular Therapeutics & New Drug Development, School of Chemistry and Molecular Engineering, East China Normal University, Shanghai 200062, China. [2] NYU-ECNU Center for Computational Chemistry at NYU Shanghai, Shanghai 200062, China. [3] Department of Chemistry, New York University, New York, NY 10003, USA. [4] Collaborative Innovation Center of Extreme Optics, Shanxi University, Taiyuan, Shanxi 030006, China. ✉email: tzhu@lps.ecnu.edu.cn; john.zhang@nyu.edu

Ever since learning to use fire, human beings have never stopped studying combustion. With increasingly serious concern on environmental pollution from combustion, understanding and mastering the combustion mechanisms is of great importance. Gaining fundamental insights into combustion processes can help us design more efficient engines and minimize the production of pollutants. A typical combustion may contain hundreds of chemical species and thousands of fundamental chemical reactions. In particular, combustion occurs at extreme physical conditions with high pressures and high temperatures up to several thousand degrees. Also, many elementary reactions in a combustion typically occur on sub picosecond time scale. These extreme physical conditions make it very difficult, if not impossible, to carry out real-time experimental study of combustion. Thus, most experimental investigations of chemical reaction mechanisms focus on individual reactions instead of the complex reaction processes occurring in a combustion. In the past decades, in slico experiments such as reactive molecular dynamics (MD) simulations have shown their values in providing molecular (atomic)-level insights into the mechanism of combustions. In a reactive MD simulation, the reaction condition can be easily controlled in the simulation and some supercritical conditions that are difficult to achieve in the experiment can also be handled. Compared with the traditional theoretical approaches such as transition sate theory and quantum collision theory that focuses on studying a single reaction, reactive MD simulation can construct the entire interwoven reaction network of a combustion system[1]. The heart of the reactive MD simulation is the potential energy surface (PES), which describes the inter- and intra-molecular interactions for molecules. Currently, there are mainly two classes of methods that can be used to construct the PES of a given molecular system: the quantum mechanics (QM)-based methods and the empirical force fields. Quantum mechanics is undoubtedly more rigorous and accurate, and MD simulations based on it are known as ab initio MD simulation (AIMD)[2,3]. Although the AIMD method in principle can simulate complex chemical reactions in real time, it is limited to relatively small systems and short simulation time (typically, dozens of picoseconds) due to exorbitant computational costs of on-the-fly ab initio calculation. With the rapid development of computer hardware and algorithms, especially the employment of graphic processing units (GPUs), some AIMD methods have recently begun to handle larger chemical systems[4]. But so far, it is still impractical to use AIMD to simulate large-scale complex reaction systems such as combustions. Over the past decades, many reactive force fields (or PESs) have been developed and successfully used for various reactive molecular systems[5–12]. A comprehensive discussion of these reactive force fields can be found in refs. [13,14]. Among these force fields, the empirical ReaxFF was widely used in MD simulation of combustion systems due to its computational efficiency[15], but its accuracy and reliability are of significant concern[16–18]. The key points of developing a reaction force field are the choice of the functional form and the parameterization process, which are complicated and depend on human intervention.

Recently, more researchers are switching to seek the help of machine-learning (ML) methods. ML method, especially artificial neural networks (NN), provides the possibility to construct PESs with the accuracy of the QM method but with an efficiency comparable to that of force fields. Neural networks constitute a very flexible and unbiased class of mathematical functions, which in principle is able to approximate any real-valued function to arbitrary accuracy. Since Behler and Parrinello proposed the high-dimensional neural network approach[19,20], several methods have been developed to implement this approach and many different kind of NN PESs have been proposed for water, small

organic molecules, and metalloid materials[21–25]. For example, the sGDML[26–28], SchNet[29], PhysNet[30], and FCHL[31] methods. NN potentials have also been employed to study the reaction mechanisms of chemical systems. By combining high-precision NN PESs and quantum collision theory, Zhang and Jiang's group have studied a series of elementary reactions in the gas phase and on the surface[32–35]. Liu and co-workers developed the LASP program to study the heterogeneous catalysis with NN PESs[36] and built stochastic surface walking (SSW)-NN to explore reaction pathways from glucose to 5-hydroxymethylfurfural[37]. Brickel et al. also studied the nucleophilic substitution reaction $[Cl–CH_3–Br]^-$ in water with NN potential[38].

In this report, we present an in silico simulation of methane combustion based on an NN potential derived by training a high-dimensional NN model from ab initio computed energies. To achieve high efficiency and accuracy, the DeePMD model was used[39–41]. This NN PES can accurately predict the energy and atomic forces of reactants, products and reaction intermediates. Based on this model, a 1-ns reactive MD simulation was performed for a combustion system initially containing 100 methane and 200 oxygen molecules with a sub-femtosecond time resolution (Fig. 1). A complete reaction network of the methane combustion can be constructed from the MD trajectory. The simulation not only produced the main reaction pathways that are consistent with the experiment but also provided much more detailed insights about the combustion processes as will be described in the following.

## Results

**Accuracy of the NN PES**. The performance of the NN potential highly depends on the quality of the reference datasets. Although several databases, such as QM7[42], QM9[43], ANI-1[44], and ANI-1x[45], are accessible, they mainly include organic molecules and are therefore not suitable for this work. Combustion of methane will generate many molecular fragments and a lot of them are free radicals[46]. Therefore, we followed a workflow (details are listed in the "Methods" section) to construct the reference datasets for the combustion. Then the DeepPot-SE model[47] was used to train the NN PES based on the reference. The predictive power of the NN model is shown in Supplementary Table 1 and Supplementary Fig. 1. It is clear that the DFT energies can be accurately reproduced by the NN model. The mean absolute errors are only 0.04 and 0.14 eV/atom in the training set and the test set, respectively. As for the atomic forces, the predicted values of the NN model are also highly consistent with the calculated results of the DFT (Supplementary Fig. 1). The correlation coefficient is 0.999 and the MAE is 0.12 eV/Å. Considering that there are a large number of atomic and molecular collisions during the combustion process, and some atomic forces can be as high as dozens of eV/Å, the accuracy of the NN model is encouraging. To verify the energy conservation of the NN PES, we performed a reactive MD simulation under the NVE ensemble. The system is a periodic box containing 100 $CH_4$ molecules and 200 $O_2$ molecules (a total of 900 atoms) with a density of 0.25 g/cm$^3$. As shown in Supplementary Fig. 2, the total energy is conserved in MD simulation.

**The initial stage of combustion**. A 1 ns reactive MD simulation was performed for methane combustion with the NN PES under the NVT ensemble. The system is also a periodic box containing 100 $CH_4$ molecules and 200 $O_2$ molecules (a total of 900 atoms) with a density of 0.25 g/cm$^3$. The MD simulations were run with a time-step of 0.1 fs and the temperature was kept at 3000 K by using the Berendsen thermostat. We chose a relatively high density (and thus high pressure) and high temperature to

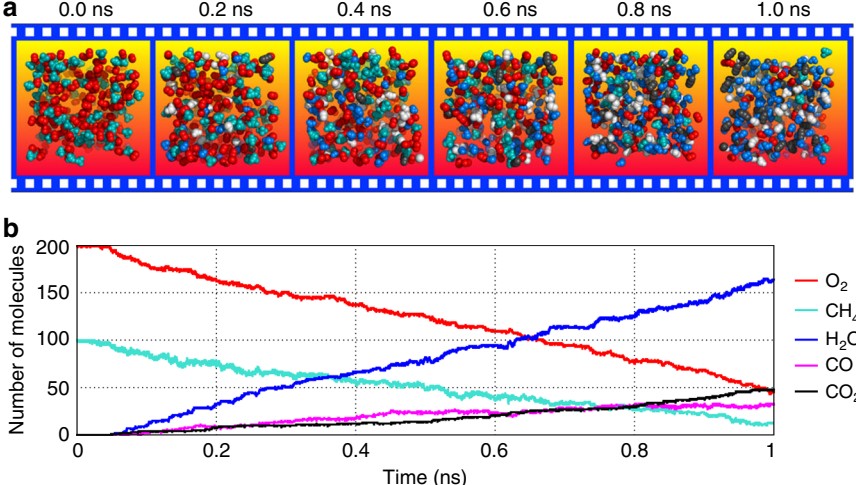

**Fig. 1 Real-time dynamics of methane combustion. a** Snapshots of the partial combustion system extracted from the reactive MD simulation of methane combustion (the time interval is 0.2 ns). The main molecular species of $CH_4$, $O_2$, $H_2O$ and $CO_2$ molecules are colored in cyan, red, blue and black, respectively. Other molecular species are colored in white. One can see that the number of reaction products were continuously increasing while reactants were being consumed. **b** Time dependences of the numbers of main molecular species in real-time MD simulation. These curves are smoothed to make them look better and clearer.

enhance the collision probability and sampling efficiency, which is a widely used strategy in reactive MD simulations because the time scale of the simulation is much shorter than that of experiments. In fact, experiments usually do not use pure fuel for combustion, but rather mix the fuel into a relatively inert gas for safety. In future work, we will try to combine the NN potential and enhanced sampling algorithms to bring simulated conditions more realistic.

Figure 1b and Supplementary Fig. 3 show the time-dependent progression of the main molecular species during the MD simulation. After 1 ns, about 90 $CH_4$ and 150 $O_2$ are consumed and about 160 $H_2O$, 30 CO, and 50 $CO_2$ are produced. The potential energy of the system during the simulation is shown in Supplementary Fig. 4. Although the system has not reached equilibrium, the important ignition process has already done, which includes much richer reaction information. In order to describe the complicated reaction network in more detail, we divided the combustion process into three stages, namely the initial stage of the combustion, the production of intermediate species of formaldehyde and formyl radical, and the production of CO and $CO_2$.

The reaction network in the initial stage of the combustion is shown in Fig. 2a. The combustion of methane started with the abstraction of its hydrogen atom by $O_2$ to generate two radicals ·$CH_3$ and HOO· (R3). As is seen from Fig. 2b, this process started at about 32 ps and took about 0.2 ps to finish. During the simulation, other radicals such as ·OH, ·H, and HOO· also abstracted hydrogen atom from $CH_4$ to generate ·$CH_3$ radical. Among them, the ·OH radical is the main species who complete this work and generates water molecules (R1). The atomization of methane into ·H and ·$CH_3$ was also observed.

Many ·$CH_3$ radicals interact with the ·OH radicals to form methanol (R6) molecules. According to Fig. 2c, this process was also very quick. Some ·$CH_3$ interacted with $O_2$ and HOO· to form methyldioxidanyl ($CH_3OO$·, R4) and methyl-hydroperoxide ($CH_3OOH$, R5). Radicals such as ·OH can also abstract H atoms from ·$CH_3$ and produce :$CH_2$. Methanol can further react with ·OH and ·H to generate methoxy radicals ($CH_3O$·, R10, R11), $H_2O$ and $H_2$. It can also react with ·H to generate ·$CH_2OH$ and $H_2$ (R12). The $CH_3O$· can also be produced by the interaction between $CH_3OO$· or $CH_3OOH$ with ·H (R8 and R9).

**Production of formaldehyde and formyl radicals**. Most methoxy radicals generated from the last step were converted to formaldehyde mainly through two reaction pathways (Fig. 3a). The first one is for methoxy radical to interact with ·OH to form formaldehyde and $H_2O$ (R16). As shown in Fig. 3b, this process took about 0.3 ps. The other pathway is for methoxy radical to interact with ·H and generate formaldehyde and $H_2$ (R17). The ·$CH_2OH$ radicals can also convert to formaldehyde by losing the hydrogen atom on its hydroxyl group (R14 and R15). If it loses one H atom on the methylene group, it can generate :CHOH radicals (R13). In addition, the :$CH_2$ radicals can interact with ·OH and form formaldehyde and the methylidyne radical (R18 and R19).

The formaldehydes were further converted into the formyl (·CHO) radicals. The main reaction pathways are hydrogen abstraction by ·O and ·OH. Figure 3c shows the trajectory of the reaction $CH_2O + ·OH \rightarrow ·CHO + H_2O$. An ·OH radical approaches the rotating formaldehyde molecule and snatches an H atom to form a water molecule; the whole process takes about 0.4 ps. In addition, other species such as ·H, $O_2$, HOO·, and ·$CH_3$ also abstracted the hydrogen atom from formaldehyde to form formyl radicals. The R20 and R23 are two reactions that form formyl radicals without the participation of formaldehyde.

**Production of CO and $CO_2$**. Formyl radicals can convert to CO by losing hydrogen in two ways (Fig. 4a). Firstly, it can lose an H atom directly (R25). Figure 4b shows a real-time trajectory of this process. A formyl radical lost its H atom at about 405.79 ps, but this reaction was quickly reversed and the formyl radical was re-formed. After another 0.4 ps the reaction took place again to form CO. Secondly, ·OH can also abstract the H atom from the formyl radical and generate $H_2O$ and CO (R26).

The formyl radical can combine with the ·OH radical to form formic acid (R24), which can further lose its H atom to form ·COOH (R27) or HCOO· (R30). These two species can convert to $CO_2$ through the reaction with ·OH or ·H (R29 and R31). The ·COOH radical can also interact with ·H and generate CO and $H_2O$ (R28). Figure 4c shows the trajectory of reaction $CO + ·OH \rightarrow CO_2 + ·H$ (R32). At 815.32 ps, an ·OH radical started to approach a CO molecule, and at 815.38 ps, an intermediate

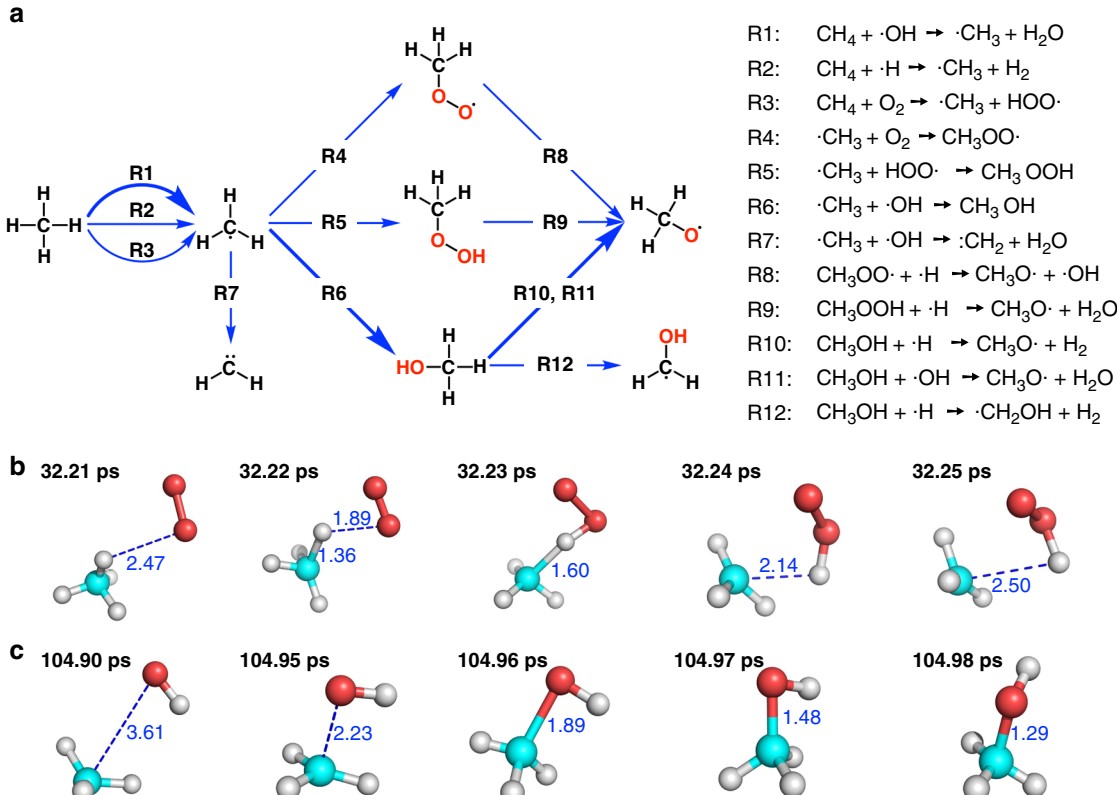

R1:   $CH_4 + \cdot OH \rightarrow \cdot CH_3 + H_2O$
R2:   $CH_4 + \cdot H \rightarrow \cdot CH_3 + H_2$
R3:   $CH_4 + O_2 \rightarrow \cdot CH_3 + HOO\cdot$
R4:   $\cdot CH_3 + O_2 \rightarrow CH_3OO\cdot$
R5:   $\cdot CH_3 + HOO\cdot \rightarrow CH_3OOH$
R6:   $\cdot CH_3 + \cdot OH \rightarrow CH_3OH$
R7:   $\cdot CH_3 + \cdot OH \rightarrow :CH_2 + H_2O$
R8:   $CH_3OO\cdot + \cdot H \rightarrow CH_3O\cdot + \cdot OH$
R9:   $CH_3OOH + \cdot H \rightarrow CH_3O\cdot + H_2O$
R10:  $CH_3OH + \cdot H \rightarrow CH_3O\cdot + H_2$
R11:  $CH_3OH + \cdot OH \rightarrow CH_3O\cdot + H_2O$
R12:  $CH_3OH + \cdot H \rightarrow \cdot CH_2OH + H_2$

**Fig. 2 The initial stage of combustion. a** Main reaction pathways in the initial stage of the combustion. **b** A real-time trajectory showing the reaction process of hydrogen abstraction from methane by $O_2$. Atoms in cyan, red and gray colors are carbon, oxygen and hydrogen, respectively. **c** A real-time trajectory showing the reaction process leading to the creation of methanol. Definition of colored atoms is the same as in (**b**).

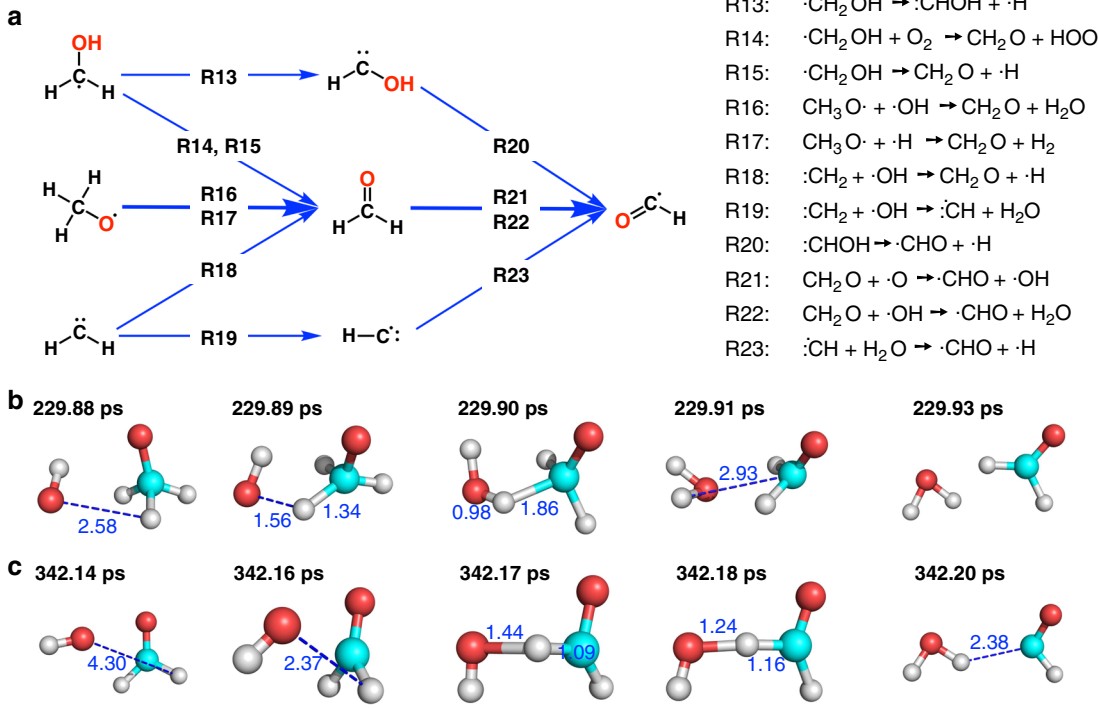

R13:  $\cdot CH_2OH \rightarrow :CHOH + \cdot H$
R14:  $\cdot CH_2OH + O_2 \rightarrow CH_2O + HOO\cdot$
R15:  $\cdot CH_2OH \rightarrow CH_2O + \cdot H$
R16:  $CH_3O\cdot + \cdot OH \rightarrow CH_2O + H_2O$
R17:  $CH_3O\cdot + \cdot H \rightarrow CH_2O + H_2$
R18:  $:CH_2 + \cdot OH \rightarrow CH_2O + \cdot H$
R19:  $:CH_2 + \cdot OH \rightarrow :CH + H_2O$
R20:  $:CHOH \rightarrow \cdot CHO + \cdot H$
R21:  $CH_2O + \cdot O \rightarrow \cdot CHO + \cdot OH$
R22:  $CH_2O + \cdot OH \rightarrow \cdot CHO + H_2O$
R23:  $:CH + H_2O \rightarrow \cdot CHO + \cdot H$

**Fig. 3 Production of formaldehyde and formyl radicals. a** The main reaction pathways for the formation of formaldehyde and formyl radicals. **b** The real-time trajectory of the reaction $CH_3O\cdot + \cdot OH \rightarrow CH_2O + H_2O$. Atoms in cyan, red and gray colors are carbon, oxygen and hydrogen, respectively. **c** The real-time trajectory of the reaction $CH_2O\cdot + \cdot OH \rightarrow \cdot CHO + H_2O$. Definition of colored atoms is the same as in (**b**).

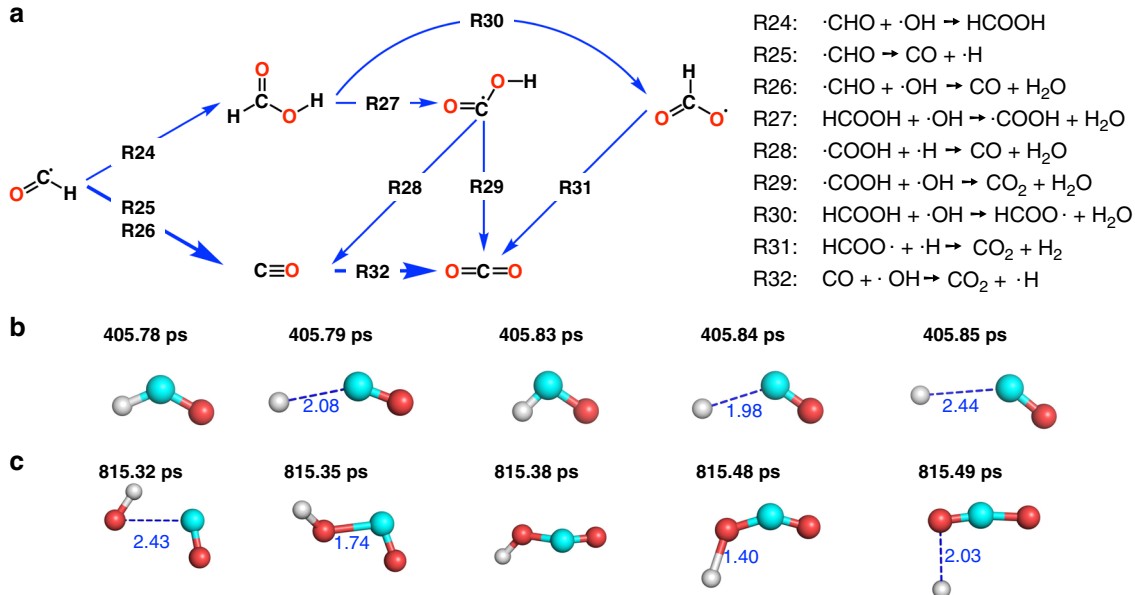

**Fig. 4 Production of CO and CO₂. a** Main reaction pathways for the formation of CO and $CO_2$. **b** The real-time trajectory of the reaction $\cdot CHO \rightarrow CO + \cdot H$. Atoms in cyan, red and gray colors are carbon, oxygen and hydrogen, respectively. **c** The real-time trajectory of the reaction $CO + \cdot OH \rightarrow CO_2 + \cdot H$. Definition of colored atoms is the same as in (**b**).

COOH was formed. The COOH should be relatively inactive, it stably existed for about 0.1 ps, and finally lost an H atom and became $CO_2$.

Further analysis found that the above-mentioned 32 reactions have all been found by experiments, and the reaction networks constructed by them are also highly consistent with the main reaction networks found experimentally[48,49]. We totally detected 505 molecular species and 798 reactions from the trajectory. Species such as ethane, ethylene, and acetylene can also be found in the experimental database. In all, 130 of the 798 reactions extracted from the MD trajectory were included in the widely accepted GRI_Mech experimental mechanism library[48]. Some experimentally observed reactions were not observed in our simulation, mostly likely because the present simulation was performed at relatively high temperature.

In fact, discovering new reactions is an important advantage of the present approach. For methane oxidation, a system that has been extensively studied by experiments, NN-based reactive MD can still discover hundreds of chemical reactions that have not been experimentally reported. This demonstrates that reactive MD can be a powerful tool to study combustion reactions. Interestingly, we found a cyclopropene molecule in the trajectory, which has not been reported to our knowledge. As shown in Supplementary Fig. 5, at 634.09 ps, a CO molecule collided with a $\cdot CH_3$ radical and joined together. Then a $CH_2CO$ molecule was formed through hydrogen loss. The $CH_2CO$ was stable for about 200 ps and then combined with another $\cdot CH_3$ radical. Subsequent hydrogen loss led to the formation of a cycloprop-2-en-1-one molecule at 828.65 ps. After another 60 ps, the third $\cdot CH_3$ attacked the cycloprop-2-en-1-one molecule and kicked out the CO group to form the $CH_3CCH_2$ molecule at 889.50 ps. Through further internal reaction and hydrogen loss, it finally formed a cyclopropene molecule at 891.16 ps and remained stable throughout the rest of the simulation. The entire process took about 260 ps to complete. While it might be possible that finding cyclopropene in our simulation is a coincidence or driven by the relatively high temperature, it still illustrates the ability of reactive MD simulation to discover new molecules and new reactions.

## Discussion

Accurate in silico MD simulation of combustion or other complex chemical reactions is one of the ultimate goals of computational chemistry. In this work, an artificial neural network potential model trained to ab initio data describes complex chemical reactions in methane combustion. This NN potential model is orders of magnitude faster than the conventional DFT calculation. Benefit from the high efficiency of the NN model and GPU acceleration, nanosecond-sale MD simulations for a chemical system containing 900 atoms was achieved in about 4 days or so on an NVIDIA Tesla P100 card. Detailed reaction mechanisms were extracted from the MD trajectory and the detected molecular species and reaction networks are in excellent agreement with experimental observation. In addition, many new reactions were found that were not included in the experimental database. Compared to laboratory experiments, in silico simulations can be performed under more extreme conditions, and any specific reaction of interest can be easily detected and tracked. In addition, MD simulation can achieve ultra-high time resolution. The time-step used in this work is 0.1 fs. With the improvement of algorithms and hardware, even resolutions in smaller time scale can be achieved.

Compared with the traditional prior knowledge-based theoretical approach, reactive MD simulation can explore complex reaction networks and discover new reactions and species without any prior knowledge of reactions. Actually, complex reactions cannot be well understood without considering the kinetics of the reaction network it belongs to. Since reactive MD simulation tracks all chemical reactions in real time, one can even deduce the rate constants for individual reactions from a single MD trajectory by statistical analysis. We extracted the ten most statistically significant reactions from the trajectory and calculated their rate constants based on the algorithms developed in previous studies[50,51]. As shown in Supplementary Table 2, most of the rate constants agree well with the GRI_Mech data[48]. The main source of error might come from the uncertainties of parameters in the Arrhenius formula and the completeness of sampling. Ideally, one should run many trajectories with different initial conditions to obtain truly statistically accurate results. However, although these

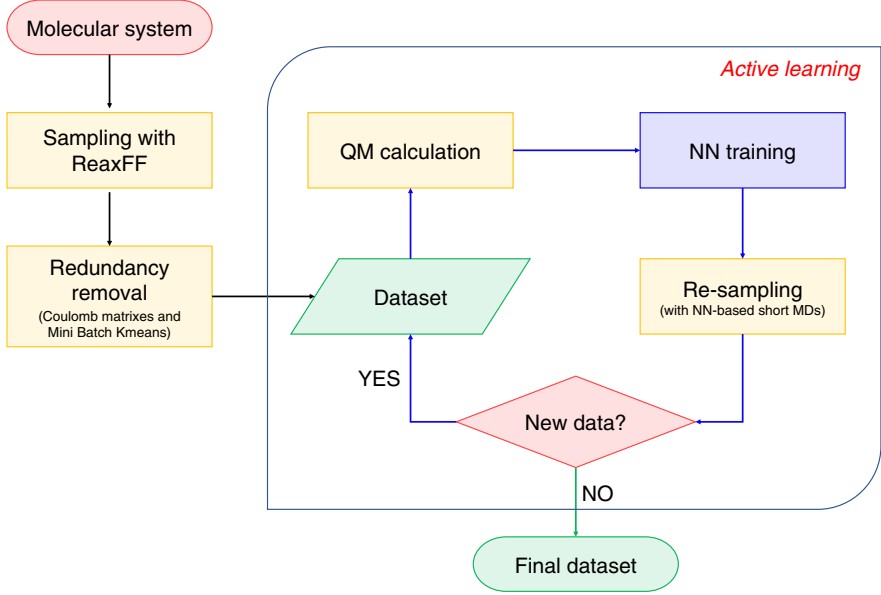

**Fig. 5 The workflow of reference dataset construction.** The process and steps used in this study to generate the reference dataset needed for neural network training to generate the potential energy for MD simulation.

rates may not be accurate enough to be used directly in kinetic modeling, they can be effective in contributing to a comprehensive understanding of the combustion reaction.

A practical issue to be pointed out is that although some algorithms were used in this study to minimize the size of the reference dataset, there are still 578,731 structures in the reference set. Although the DFT calculation is very efficient, such a large reference set is difficult to perform high-level post-Hartree−Fock calculations. In order to further minimize the size of the reference set while ensuring its completeness, new algorithms need to be developed to further enhance the efficiency of this approach. Recently, Zhang et al. developed the DP-GEN[52] (Deep potential Generator) software platform, which can automatically construct the reference dataset and train the NN model. The concurrent learning algorithm employed by this platform can make the redundancy of the reference set as small as possible. We are trying to integrate the algorithms developed in this work into the DP-GEN platform.

In addition, it is worth to point that while combustion is usually thought to be dominated by free radical reactions, recent studies have begun to examine the role of electronically excited state species in combustion. For example, the additional introduction of plasma was found to be effective in promoting combustion in experiments[53]. However, MD simulations involving excited states are highly nontrivial, and there are large uncertainties in ab initio quantum chemistry computation for treating excited states of large systems. Based on sophisticated empirical or machine-learning PESs, several recent works have achieved the excited-state MD simulation for model systems[54–62]. For example, the O+O recombination reaction to form the ground and excited-state singlet $O_2$ molecules on amorphous solid water[60]. Such strategy will be considered in our future studies.

Despite further improvement is needed, the current report heralds the dawn of a new era in which neural network-based reactive MD simulation can be practically applied to simulating complex reaction systems at the ab initio level, which provides atomic-level understanding of every reaction process at unprecedented level of details beyond what laboratory experiment can accomplish.

## Methods

**Reference dataset**. In this study, a workflow was developed for making reference datasets (Fig. 5). The details of each module in the workflow are given below.

To increase the efficiency of dataset construction, reactive MD simulation with ReaxFF was used to sample an initial dataset. A model combustion system containing a lot of $CH_4$ and $H_2$ molecules was built by using the Amorphous Cell module in the Material Studio[63] software package. Then the LAMMPS[64] program was used to perform the MD simulation. The NVT ensemble was used and the temperature was set to 3000 K with the Berendsen thermostat. The ReaxFF parameter of Chenoweth et al. (CHO-2008 parameter set)[65] was employed. The Open Babel software[66] and the Depth-First Search algorithm[67] were used to detect species in every snapshot of the trajectory. Then, for each atom in each snapshot, we build a molecular cluster that contains this atom and species that within a specified cutoff centered on it. In this work, the cutoff was set to 5 Å.

The initial dataset contains about 22.5 million structures, which is too large to perform QM calculations for every molecular cluster it contains. Therefore, it is necessary to resample it to remove redundant structures while ensuring its completeness. To this end, we first classified the initial dataset into sub-datasets based on the chemical bond information of the central atom. For example, the central H atom can be classified into two different types: a single H atom (H0) and an H atom formed a single chemical bond with another atom (H1).

Further treatment is still needed for large sub-datasets. For a given large sub-dataset, we first expressed each molecular cluster it contains as a Coulomb matrix[68]:

$$\mathbf{C}_{ij} = \begin{cases} \frac{1}{2} Z_i^{2.4}, i = j \\ \frac{Z_i Z_j}{|\mathbf{R}_i - \mathbf{R}_j|}, i \neq j \end{cases}, \quad (1)$$

where $Z_i$ and $Z_j$ are nuclear charges of atom $i$ and $j$, $\mathbf{R}_i$ and $\mathbf{R}_j$ are their Cartesian coordinates. The minimum image convention[69] was used to consider the periodic boundary condition. "Invisible atoms" were introduced to fix the dimension of the Coulomb matrix. These invisible atoms do not influence the physics of the molecule of interest and make the total number of atoms in the molecule sum to a constant. To lower the dimension of the dataset and keep as much structural information as possible, the Coulomb matrix was further represented by the eigen-spectrum, which is obtained by solving the eigenvalue problem $\mathbf{C}\mathbf{v} = \lambda\mathbf{v}$ under the constraint $\lambda_i \geq \lambda_{i+1}$. The clustering algorithm Mini Batch KMeans[70] was then used to cluster the given sub-datasets into smaller clusters according to the eigen-spectrum. Then we randomly selected 10,000 structures from each cluster (If the cluster contains no more than 10,000 structures, then all of them were selected).

Large amplitude collisions and reactions in the combustion can produce a lot of unpredictable species and intermediates. To ensure the completeness of the reference dataset, an active learning approach[71] was used. Four different NN PES models were trained based on the dataset from the last step. Then several short MD simulations were performed based on these NN models. During the simulation, the atomic forces are evaluated by these four NN PES models simultaneously. For a

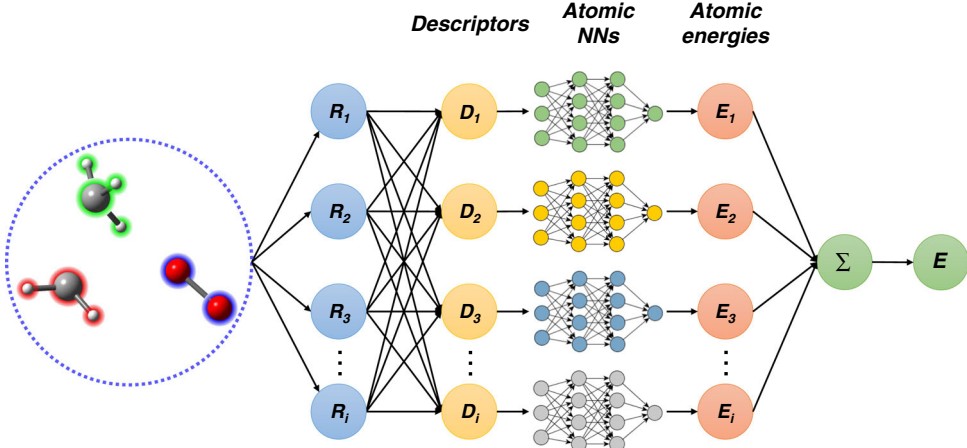

**Fig. 6 The neural network model.** The neural network model that generates the potential energy surface for MD simulation.

specific atom, if the predicted forces by these four models are consistent with each other, then the molecular cluster that centered on this atom should be found in the dataset. On the contrary, if the results of these four models are inconsistent with each other and the error between them is in a specific range (0.5 eV/Å < error < 1.0 eV/Å in this work), the corresponding molecular cluster will be added into the dataset. The update of the dataset will be continued until the predictions of the four models are always consistent.

**QM calculation**. The potential energy and atomic forces for every structure in the final dataset were calculated by Gaussian 16[72] software at the MN15/6-31G** level. The MN15 functional was employed because it has broad accuracy for multi-reference and single-reference systems[73]. To consider the spin polarization effect, the initial wave function of a given structure is obtained by the combination of the wave functions of individual molecular species forming the structure, while the wave function of each molecular species was calculated based on its own charge and spin.

**Training of the NN PES**. The scheme of the NN model is shown in Fig. 6. The total energy $E$ of a given structure is decomposed into a sum of atomic energy contributions[19,74], i.e., $E = \sum_i E_i$, where $i$ is the index of the atom. Each atomic energy is fully determined by the position of the $i$th atom and its near neighbors. To guarantee the translational, rotational, and permutational symmetries lying in the PES, the Cartesian coordinates of atomics are mapped to specific mathematical formulas called "descriptors" of the atomic chemical environment.

The DeepPot-SE (Deep Potential-Smooth Edition) model[47] was used to train the NN potential by the DeePMD-kit program[74]. Details of this method can be found in ref. [67]. The model includes two networks: the embedding network and the fitting network. Both networks use the ResNet architecture[75]. The size of the embedding network was set to (25, 50, 100) and the size of the embedding matrix was set to 12. The size of the fitting network is set to (240, 240, 240). The cutoff radius was set to 6.0 Å and the descriptors decay smoothly from 1.0 to 6.0 Å. The initial learning rate was set to 0.0005 and it will decay every 20,000 steps. The loss is defined by

$$\mathcal{L} = \frac{p_e}{N}\Delta E^2 + \frac{p_f}{3N}\sum_i |\Delta \mathbf{F}_i|^2, \qquad (2)$$

where $\Delta E$ and $\Delta \mathbf{F}_i$ are root mean square errors in energy and force. The prefactor $p_e$ is set to 0.2 eV$^{-2}$ and the $p_f$ decays from 1000 Å$^2$ eV$^{-2}$ to 1 Å$^2$ eV$^{-2}$.

## Data availability
The datasets (structures, potential energies and atomic forces of molecular species) generated during the current study are available at https://github.com/tongzhugroup/NNREAX, https://doi.org/10.6084/m9.figshare.12973055. Source data are provided with this paper.

## Code availability
The codes used to generate the datasets in the current study are available at https://github.com/tongzhugroup/mddatasetbuilder, https://doi.org/10.5281/zenodo.4035925.

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

## Acknowledgements

The authors thank Dr. Linfeng Zhang and Dr. Han Wang for their discussion and help in using DeePot-SE and DeePMD-kit. T.Z. would also like to thank Prof. Donghui Zhang for his valuable suggestions in this project. This work was supported by the National Key R&D Program of China (grant no. 2016YFA0501700), the National Natural Science Foundation of China (grant nos. 91641116, 91753103, and 21933010), and the Innovation Program of Shanghai Municipal Education Commission (201701070005E00020). J. Zeng was partially supported by the National Innovation and Entrepreneurship Training Program for Undergraduate (201910269080). We also thank the ECNU Multifunctional Platform for Innovation (No. 001) for providing supercomputer time.

## Author contributions

J.Z. trained the neural network potential and performed most of the QM calculations. L.C. and M.X. analyzed the trajectory and performed part of the QM calculation. T.Z. and J.Z.H.Z. conceived the project and wrote the manuscript with input from all authors.

## Competing interests

The authors declare no competing interests.
