## [Peer Review File · Nature Communications]

REVIEWER COMMENTS

Reviewer #1 (Remarks to the Author):

This manuscript presents an neural network-based investigation of the combustion of methane. Using NN-based energy functions is an approach that receives considerable attention and may provide a more complete understanding of complex chemical processes. This is an interesting application to a challenging problem. What is not yet obvious is how quantitative the simulations are and how to best test, ascertain and validate this.

Detailed comments:

1. The reference calculations were carried out at the DFT level of theory. However, combustion processes involve high temperature, high pressure, or both with the possibility to populate excited electronic states. Is this taken into account? For recent work on this see Marquetand et al. <https://arxiv.org/abs/1912.08484>

Also, at these conditions multiconfigurational effects will be important. A broader discussion of the expected uncertainties from using MN15 compared with more rigorous methods, such as MRCI+Q, is warranted.

2. Considering the rates in Table S2 it is noted that for reactions involving production of OH two of them ($O + CH_4$ and $H + H_2O$) agree very favourably with experiment but for the third ($CH_3 + H_2O$) it differs by almost two orders of magnitude. Is there an explanation? Also, are the reaction conditions under which the experimental rates have been recorded comparable to those encountered in the simulations? Often, such experiments are carried out under relatively controlled conditions whereas here there are many spectator species present.

3. There are other modern approaches to devise reactive force fields that can be used in dynamics simulations which should be cited. One of them uses permutationally invariant polynomials (Bowman et al., *Intern. Rev. Phys. Chem.* 2009, 28, 577–606); another one is adiabatic reactive MD (Nagy et al., *J. Chem. Theory Comput.* 2014, 10, 1366–1375), and a third one is EVB pioneered by Warshel.

4. The focus in the present work appears to be more on computability and less so on accuracy (e.g. level of theory of quantum chemistry, see point 1). Although the performance of the model has been assessed (Figure S3, Table S1), the performance is not comparable to other

approaches such as SchNet, PhysNet (Unke et al., J. Chem. Theory Comput. 2019, 15, 3678– 3693, which should be cited), or FCHL (Christensen et al., JCP 152, 044107 (2020), which should be cited, too). Such NNs train routinely to MAE well below 1 kcal/mol on QM9. Hence, the question on the present work is how reliable the computations and the information derived from them are.

5. Concerning the simulation conditions - are pressure, temperature, and number densities representative for combustion?

6. One of the problems with other approaches in the field is total energy conservation. Is total energy conserved here? Demonstration of this by running an MD simulation without thermostat will be essential.

7. In Figure 1: does the labeling in Panel B mean that a time interval from 0 to 1 ns is shown? Maybe it would be better to have these labels on the bottom of the Figure? Also, does the system reach equilibrium here? If not, it may be useful to mention this.

8. It will be useful to have the same Figure as S3 for energies to see in particular the spread around the average.

9. The authors mention specifically that the approach can be used for discovering new reactions on p. 9. What do they mean when writing "...we found a cyclopropane...which has not been reported to our knowledge.."? That in the combustion of CH₄ by O₂ formation of this product is novel? I would have intuitively assumed that this is not particularly surprising given the simulations conditions. The authors should elaborate more on why this is surprising.

In summary, the present work is a potentially interesting application of molecular dynamics together with machine learned energy functions. What remains unclear is how accurate and quantitatively meaningful the present approach is given the relatively low level of theory used for the reference data. Combustion processes can involve electronically excited states and may require higher level methods and careful assessment of the present model in these regards is warranted before the manuscript can be further considered for publication in this journal.

Reviewer #2 (Remarks to the Author):

This is a nice paper that studies the reaction dynamics of realistic combustion systems using molecular dynamics with the state-of-art machine learning based potential energy surface (PES). The authors were able to conduct nano-second simulation on a reasonably large system, with quantum accuracy.

These kinds of calculations were not possible before.

As such, this is an important contribution to the study of combustion.

At a more detailed level, the authors were able to construct the reaction network using the molecular dynamics simulation results.

The paper lacked quantitative results, such as reaction rates, that can be calibrated against other results, e.g. experimental results. This is perhaps the one shortcoming that stands out the most.

The writing of the paper can also be improved. The introduction reads more like a laundry list. This paper is not a review of machine learning based PES. So I suggest that the authors focus on what they are using, instead of trying to list all other possibilities.

The data generation part seems a bit ad hoc. The accuracy of the model relies heavily on data. So this part is crucial. There are some rather systematic data generation procedures discussed in the literature in connection with machine learning based PES. The authors might want to look into those.

Reviewer #1

This manuscript presents a neural network-based investigation of the combustion of methane. Using NN-based energy functions is an approach that receives considerable attention and may provide a more complete understanding of complex chemical processes. This is an interesting application to a challenging problem. What is not yet obvious is how quantitative the simulations are and how to best test, ascertain and validate this.

Detailed comments:

- 1. The reference calculations were carried out at the DFT level of theory. However, combustion processes involve high temperature, high pressure, or both with the possibility to populate excited electronic states. Is this taken into account? For recent work on this see Marquetand et al. <https://arxiv.org/abs/1912.08484>. Also, at these conditions multiconfigurational effects will be important. A broader discussion of the expected uncertainties from using MN15 compared with more rigorous methods, such as MRCI+Q, is warranted.*

Reply: Thank you very much for your comments and suggestions. We would like to address the above question in the following.

A. Combustion is a complex reaction, even a system as simple as methane oxidation contains hundreds of species and reactions. The complexity can increase even further when the changes of starting configuration (e.g., the stoichiometric ratio of reactants) and the external conditions such as temperature are considered. Researches such as the design of engines and fuels require a comprehensive understanding of the combustion mechanism. However, sophisticated studies of individual reactions cannot provide this information effectively. Thus, the current work mainly focuses on developing a computational protocol for the exploration and construction of complete reaction networks in combustion rather than rigorous studies of any individual chemical reactions.

B. Electronic excitation can be important for some individual chemical reactions. However, for combustion reactions at high pressures, there are rapid energy exchanges between molecules resulting from high-frequency collisions, in which free radicals are considered to dominate combustion reactions (see, e.g., *Cleaner Combustion Developing Detailed Chemical Kinetic Models*, Springer, 2013) and the effect of electronic excitation tend to be less important. For example, most theoretical computations for combustion reactions do not consider excited electronic states, and the calculated reaction rates agree well with experiment measurements (e.g., *Combustion and Flame* 200, 125, 2019.; *Combustion and Flame*, 197, 423, 2018.; *Energy Fuels*, 34, 949, 2020.). Currently, widely used databases of small molecule combustion mechanisms such as *GRI_Mech* and *AramcoMech* also did not include excited-state species.

C. A typical reactive MD simulation contains thousands of atoms and involves hundreds or even thousands of chemical reactions and intermediate species. Current *ab initio* methods are generally unable to accurately calculate electronically excited states of chemical reactions for such complex reaction systems.

D. Molecular dynamics simulations involving excited states are highly non-trivial. And there are large uncertainties in *ab initio* quantum chemistry computation for reaction rate constants involving excited species and electrons (see, e.g., *Progress in Energy and Combustion Science*, 48, 21, 2015.).

E. Theoretical treatment of nonadiabatic dynamics using empirical approaches is still a great challenge, and currently these methods are still controversial in their applications.

We will explore the methods developed by Prof. Marquetand and co-workers (<https://arxiv.org/abs/1912.08484>, *Mach. Learn.: Sci. Technol.* 1, 025009 2020) for possible application in combustion involving excited species and a comment on this aspect was added in the revised manuscript:

“In addition, it is worth to point that while combustion is usually thought to be dominated by free radical reactions, recent studies have begun to examine the role of electronically excited state species in combustion. For example, the additional introduction of plasma was found to be effective in promoting combustion in experiments⁵¹. However, molecular dynamics simulations involving excited states are highly non-trivial. Based on artificial neural network models, several recent pioneering works have achieved the excited-state MD simulation for model systems⁵²⁻⁵⁵. We will consider the introduction of these algorithms into the simulation of combustion reactions in future work.”

Referring to the accuracy of the MN15 DFT method used in the current study, the MN15 functional was chosen because it is specifically designed to have broad accuracy for multi-reference and single-reference systems. When compared with 82 other density functionals, MN15 gives the second smallest mean unsigned error (MUE) for 54 inherently multiconfigurational systems (*Chem. Sci.*, 7, 5032, 2016.). The MN15’s training set contains the bond dissociation enthalpies of small organic molecules and these data have been calibrated by experimental measurements and high-level *ab initio* multireference methods (e.g. *J. Phys. Chem. A* 120, 4025, 2016.).

Also, to cover the chemical space of the combustion reaction, the training set in our work is very large, and the active learning method was used to automatically expand the training set. This means that the MRCI+Q method, which is computationally intensive and requires expert experience (especially the choice of active space), is not computationally practical in our current study.

In addition, most of QM software do not provide analytical gradient at the MRCI+Q level, but the gradient is crucial for training the neural-network potential energy surface.

To test the accuracy of MN15 against MRCI+Q, we calculated the bond dissociation energy curves of the C-C and C-H bonds in ethane using both the MRCI+Q (8 active electrons and 8 active spaces.) and the MN15 methods, with the same 6-31G** basis set. The result (see figure below) shows that MN15 and MRCI+Q are in good agreement with each other.

Fig. 1. Bond dissociation energy curves of the C-C and C-H bonds in ethane calculated by MN15 and MRCI+Q methods.

2. Considering the rates in Table S2 it is noted that for reactions involving production of OH two of them ($O + CH_4$ and $H + H_2O$) agree very favorably with experiment but for the third ($CH_3 + H_2O$) it differs by almost two orders of magnitude. Is there an explanation? Also, are the reaction conditions under which the experimental rates have been recorded comparable to those encountered in the simulations? Often, such experiments are carried out under relatively controlled conditions whereas here there are many spectator species presents.

Reply: Thank you for the comment.

The experimental rates are not obtained from a single experimental source, but from fitting multiple sets of experimental data to the Arrhenius formula, which means that these values are averaged rates over many different experiments and thus have some uncertainties. Another source of error is the completeness of the sampling. For those reactions that appear less frequently in our MD simulation, the calculation of reaction rates from a single trajectory can cause large errors. Ideally, one should run many trajectories with different initial conditions to obtain statistically more accurate rate constants.

Usually, for the combustion reaction, it is already considered a good match when the difference between the predicted and experimental rates is around an order of magnitude. In fact, most rates calculated in this work are comparable in accuracy to other fast rate calculation methods based on statistical or machine learning approaches (see, i.e., *Communications in Information and Systems*, 19, 4, 2019.; *J. Comput. Chem.* 40, 1586, 2019.).

More importantly, the main advantage of the current work is not in the precise calculation of the individual reaction rate. However, by MD simulation, one can extract the reaction rates of many important reactions from the trajectory. Some of these rates might not be accurate enough to be used directly in kinetics modeling, but they can be highly useful in our overall understanding of the combustion reaction.

These comments and explanations were added in the revised manuscript:

“The main source of error might come from the uncertainties of parameters in the Arrhenius formula and the completeness of sampling. Ideally, one should run many trajectories with different initial conditions to obtain truly statistically accurate results. However, although these rates may not be accurate enough to be used directly in kinetic modeling, they can be effective in contributing to a comprehensive understanding of the combustion reaction.”

3. *There are other modern approaches to devise reactive force fields that can be used in dynamics simulations which should be cited. One of them uses permutationally invariant polynomials (Bowman et al., Intern. Rev. Phys. Chem. 2009, 28, 577–606); another one is adiabatic reactive MD (Nagy et al., J. Chem. Theory Comput. 2014, 10, 1366–1375), and a third one is EVB pioneered by Warshel.*

Reply: Thank you very much. We are sorry for our mistake. These works have been cited in the revised manuscript.

4. *The focus in the present work appears to be more on computability and less so on accuracy (e.g. level of theory of quantum chemistry, see point 1). Although the performance of the model has been assessed (Figure S3, Table S1), the performance is not comparable to other approaches such as SchNet, PhysNet (Unke et al., J. Chem. Theory Comput. 2019, 15, 3678–3693, which should be cited), or FCHL (Christensen et al., JCP 152, 044107 (2020), which should be cited, too). Such NNs train routinely to MAE well below 1 kcal/mol on QM9. Hence, the question on the present work is how reliable the computations and the information derived from them are.*

Reply: Thank you for your suggestion, these works have been cited in the revised manuscript.

However, the comparison in accuracy between our work and the above works should be viewed in perspective. This is because the QM9 dataset used in those works mainly contains small organic molecules that were energy minimized and the related works mainly focused on the energies of stable molecular species, not chemical reactions. While in our present work, we need to deal with the breaking and formation of chemical bonds, which are accompanied by much greater changes of energy and configurational space than those of stable molecular species. In fact, considering such a large range of changes in energy and force, the accuracy of the current study is already high. For example, the MAEs on the training and test sets are 0.94 and 3.22 kcal/mol, respectively. And the performance of the current model is comparable to other models that were trained on datasets with similar value ranges. (see, e.g., *Chem. Sci.*, 2019, 10, 8100).

In addition, it should be pointed out that the reaction rates determined by the Arrhenius equation at high temperatures are not as sensitive to the accuracy of potential energy surface as those at low temperatures. Thus, the overall reaction networks generated from the MD simulation of combustion at high temperature shall not be seriously affected by the small errors in potential energy.

In summary, we believe that the accuracy of our study is quite reliable in general.

5. *Concerning the simulation conditions - are pressure, temperature, and number densities representative for combustion?*

Reply: Compared to the methane combustion experiments, we increased the density (and thus the pressure) as well as the temperature to enhance the collision probability and sampling efficiency. Detailed values are provided in the manuscript. Increasing the reactant density and temperature are widely used strategies in reactive MD simulation because the time scale of the simulation is much shorter than that of experiments. In future work, we will try to combine neural network potential and enhanced sampling algorithms to bring the simulated conditions closer to the experiments. These comments were added in the revised manuscript.

“Compared to the experiments, we increased the density (and thus the pressure) as well as the temperature to enhance the collision probability and sampling efficiency, which are widely used strategies in reactive MD simulation because the time scale of the simulation is much shorter than that of experiments. In future work, we will try to combine the NN potential and enhanced sampling algorithms to bring the simulated conditions closer to the experiments.”

6. *One of the problems with other approaches in the field is total energy conservation. Is total energy conserved here? Demonstration of this by running an MD simulation without thermostat will be essential.*

Reply: The total energy of the current model is conserved. We have trained system to predict both potential energy and its gradient, which means that the second-order derivatives of the potential energy surface are continuous. In the following, we show a 10ps NVE simulation result using the current model starting with a random frame extracted from the trajectory. As shown in the following figure, the total energy is conserved, only with little numerical errors. We didn't perform a longer simulation because combustion is an exothermic reaction, thus the temperature of the system rises sharply in the NVE ensemble, soon exceeding the chemical space covered by the current training set.

Fig. 2. The total energy of the system over time under the NVE ensemble. The total energy of the first snapshot (-4632920.5eV) was taken as the reference value.

7. *In Figure 1: does the labeling in Panel B mean that a time interval from 0 to 1 ns is shown? Maybe it would be better to have these labels on the bottom of the Figure? Also, does the system reach equilibrium here? If not, it may be useful to mention this.*

Reply: Thank you very much for your suggestion. We have modified Figure 1 according to this suggestion.

After 1 ns of simulation, the system did not reach equilibrium. We didn't extend the simulation because in reactive MD simulations of combustion, the trajectory before equilibrium is often more important because it contains the ignition process and can provide richer reaction information. Whereas only oscillatory reactions between products are usually observed in the post-equilibrium trajectory.

More than 75% of the methane and oxygen have been consumed and more than 150 Water molecules and lots of carbon dioxide and carbon monoxide have been produced in the 1ns simulation. Therefore, we think it's reasonable to analyze the trajectory of the period.

We added a comment in the revised manuscript:

“The potential energy of the system during the simulation is shown in Fig. S3. Although the system hasn't reach equilibrium, the trajectory already contains the ignition process and can provide richer reaction information.”

8. *It will be useful to have the same Figure as S3 for energies to see in particular the spread around the average.*

Reply: Thank you for this suggestion. A new figure which shows the change of potential energy during the simulation was added in the revised manuscript.

9. *The authors mention specifically that the approach can be used for discovering new reactions on p. 9. What do they mean when writing ". we found a cyclopropane...which has not been reported to our knowledge.."? That in the combustion of CH₄ by O₂ formation of this product is novel? I would have intuitively assumed that this is not particularly surprising given the simulations conditions. The authors should elaborate more on why this is surprising.*

Reply: We are sorry that we didn't make our point clearer. What we want to express is that finding new reactions is an important advantage of the present approach. For methane oxidation, a system that has been extensively studied by experiments, NN based reactive MD can still find hundreds of chemical reactions that have not been experimentally reported. This demonstrates that reactive MD can be a powerful tool to study combustion reactions.

In previous works by Martinez and co-workers (*Nature Chemistry* 6, 1044,2014; *ACS Cent. Sci.* 5, 1532, 2019.), GPU based AIMD simulations were used to simulate the Urey-Miller experiment at the HF/3-21G level and a synthetic pathway of glycine was discovered.

We believe that NN based reactive MD simulation is an efficient and accurate tool to find new molecules and chemical reactions. The synthetic pathway of cyclopropane found in our MD simulation was just as an example. In fact, although there has been a lot of research into methane combustion, including the construction of several reaction databases, there are no experimental reports on cyclopropane generated from combustion. While it might be possible that finding cyclopropane in our simulation is a coincidence, it still illustrates the ability of reactive MD simulation to discover new molecules and new reactions.

We have modified this description in the revised manuscript to avoid ambiguity.

Reviewer #2

This is a nice paper that studies the reaction dynamics of realistic combustion systems using molecular dynamics with the state-of-art machine learning based potential energy surface (PES). The authors were able to conduct nano-second simulation on a reasonably large system, with quantum accuracy.

These kinds of calculations were not possible before. As such, this is an important contribution to the study of combustion.

At a more detailed level, the authors were able to construct the reaction network using the molecular dynamics simulation results.

1. The paper lacked quantitative results, such as reaction rates, that can be calibrated against other results, e.g. experimental results. This is perhaps the one shortcoming that stands out the most.

Reply: Thank you very much for your comments.

We extracted the 10 most statistically significant reactions from the trajectory and calculated their rate constants (Table S2). Most of these rate constants agree well with the experiments and are comparable in accuracy to other fast rate calculation methods based on statistics or machine learning (see, i.e., *Communications in Information and Systems*, 19, 4, 2019.; *J. Comput. Chem.* 40, 1586, 2019.).

2. The writing of the paper can also be improved. The introduction reads more like a laundry list. This paper is not a review of machine learning based PES. So I suggest that the authors focus on what they are using, instead of trying to list all other possibilities.

Reply: Thank you very much. The manuscript was revised according to this suggestion.

3. The data generation part seems a bit ad hoc. The accuracy of the model relies heavily on data. So this part is crucial. There are some rather systematic data generation procedures discussed in the literature in connection with machine learning based PES. The authors might want to look into those.

Reply: Thank you very much for this suggestion.

In the current work, we have used the active learning algorithm to automatically expand the training set. In order to generalize this method for all hydrocarbon fuels and enhance its user-friendliness, we are trying to combine this method with the DP-GEN software which can

greatly simplify the preparation process of the dataset. We are also trying to make a basic reference data set for all hydrocarbon fuels.

Relevant works are ongoing in our lab.

Further clarification on this issue was added in the revised manuscript:

“Recently, Zhang et al. developed the DP-GEN50 (Deep potential Generator) software platform which can automatically construct the reference dataset and train the NN model. The concurrent learning algorithm employed by this platform can make the redundancy of the reference set as small as possible. We are trying to integrate the algorithms developed in this work into the DP-GEN platform.”

REVIEWER COMMENTS

Reviewer #1 (Remarks to the Author):

The authors have addressed a number of points. However, several points remain to be clarified.

1. In the introduction the authors write that "Although empirical reactive force fields[.]however, their accuracy and reliability are of significant concern." It should be noted that the reactive force fields by Bowman and coworkers, or Meuwly and coworkers are usually within 1 kcal/mol or even better of the reference data which is typically at the MP2 or more often MRCI or CCSD(T) levels of theory. So this statement clearly needs context and not all reactive force fields should be discussed in this overarching fashion.
2. While pointing out the inaccuracies of some existing reactive force fields may be justified, the authors themselves then "excuse" their treatment of the systems at the DFT level by pointing out in their reply letter that "..the current work mainly focuses on developing a computational protocol for the exploration...rather than rigorous studies of any individual chemical reactions." This raises the question whether by "inaccurately" describing individual chemical reactions much insight into the complex reaction network can be gained they consider. Again, this point needs considerably more careful context.
3. While electronic excitation per se may not be that common, reactive recombination to form excited states is clearly a possibility. This should be mentioned. One example has been recently given for O+O recombination (Pezzella et al. (2020)) in which ground and excited state O₂ was formed upon recombination. As the reactivity of electronically excited O₂ can be several times larger towards carbo-hydrates, including electronically excited state in this example is clearly essential. What is the situation in the present work?
4. Again, in their reply the authors point out the limitations of current electronic structure methods for treating excited states of large numbers of systems. This should be mentioned more explicitly in the manuscript. There are actually recent reactive MD studies of nonadiabatic processes involving several electronic states, see Koner et al., Varandas et al., Schinke et al. and others, all on high-quality PESs.
5. Point 2 from reviewer 1 is only incompletely answered. How different are the reaction conditions for the reactions generating OH from those used in the present simulations?
6. The authors refer to Arrhenius forms for the rate coefficients. In hypersonics, often a modified Arrhenius with an explicitly T-dependent prefactor is used. As this manuscript deals with combustion, such modified forms might offer some advantage and should be mentioned and discussed.
7. The reply to point 5, reviewer 1 is rather vague. The authors

should state what density was used instead of reporting "...increased [the] density.." and similar for all other system variables. A short discussion of the differences between experimental reaction conditions and those used in the simulations is clearly necessary.

8. The figure from their reply letter on the energy conservation should be included at least in the SI. A larger scale for the energy has to be given and the simulation time needs to be considerably extended. Drifts in the total energy can not be seen from a 10 ps time series. Would the simulation leading to Figure 1 be suitable for that?

9. While it is certainly true that finding new reactions is an attractive possibility of such a computational model, it should be stated explicitly that the accuracy of the model determines whether or not conclusions about how likely and feasible such new reactions are valid. To validate this, one such "new reaction" - e.g. the one forming cyclopropane - that was found must be investigated at a higher level of theory. From the reaction profile it should then be determined whether or not "finding this new reaction" is an artifact of the model because its accuracy is not appropriate or whether the "new reaction" is indeed feasible at the reaction conditions chosen in the simulations. Only then the conclusion from the reply letter "...demonstrates that reactive MD can be a powerful tool to study combustion reactions." is likely to be valid.

10. In Figure 1 it is interesting to note that all concentrations are either monotonically increasing or decreasing. Is there an explanation for this?

Minor point: the formatting of the references needs attention, e.g. Ref.33 has no page numbers, ditto for Ref. 16.

In summary, this work is a potentially useful way forward to investigate complex reactions in combustion. However, at several places the accuracy of the model needs to be more critically scrutinized, established, or demonstrated. Without critical validation it remains unclear whether or not the conclusions are sufficiently supported.

Reviewer #1

The authors have addressed a number of points. However, several points remain to be clarified.

1. In the introduction the authors write that "Although empirical reactive force fields[..]however, their accuracy and reliability are of significant concern." It should be noted that the reactive force fields by Bowman and coworkers, or Meuwly and coworkers are usually within 1 kcal/mol or even better of the reference data which is typically at the MP2 or more often MRCI or CCSD(T) levels of theory. So this statement clearly needs context and not all reactive force fields should be discussed in this overarching fashion.

Reply: Thank you for this comment. By “empirical reactive force fields”, we mean the ReaxFF force field and does not include accurate potential energy surfaces (PES) of Bowman, Meuwly, etc. The ReaxFF is widely used in the simulation of combustion while those more accurate PES haven't been designed for combustion simulations of large molecular systems (*WIREs Comput Mol Sci.* 2019;9:e1386.). While the accuracy and reliability are of significant concern. In a recent work by Head-Gordon and co-workers (*J. Phys. Chem. A* 2020, 124, 27, 5631.), the performance of several common parametrizations of the ReaxFF force field against DFT and CCSD(T) were benchmarked. The results show that the ReaxFF potentials fail both quantitatively and qualitatively to describe reactive events relevant to hydrogen combustion systems, while the DFT results show good agreement with that of CCSD(T).

We've revised the manuscript accordingly to avoid ambiguity.

2. While pointing out the inaccuracies of some existing reactive force fields may be justified, the authors themselves then "excuse" their treatment of the systems at the DFT level by pointing out in their reply letter that "...the current work mainly focuses on developing a computational protocol for the exploration...rather than rigorous studies of any individual chemical reactions." This raises the question whether by "inaccurately" describing individual chemical reactions much insight into the complex reaction network can be gained they consider. Again, this point needs considerably more careful context.

Reply: Thank you very much for your suggestion. But this is not what we tried to express.

DFT is still the most widely used quantum chemistry method for studying chemical reactions and its accuracy has been widely accepted by the quantum chemistry community. Although DFT is in general not considered as accurate as more rigorous methods such as

MRCI or CCSD(T), but these more accurate methods are currently not suitable for studying complex chemical reactions of large systems such as the combustion process in the current work.

Also, DFT methods often produce results that are comparable in accuracy to these more accurate results but at a fraction of the computational cost. For example, we demonstrated that our DFT calculation (with the MN15 functional) for model reaction systems is comparable in accuracy to MRCI+Q and CCSD(T) (Fig 1 and 3 in this response). The MN15 functional was chosen in our calculation because it is specifically designed to have broad accuracy for multi-reference and single-reference systems. When compared with 82 other density functionals, MN15 gives the second smallest mean unsigned error (MUE) for 54 inherently multiconfigurational systems (*Chem. Sci.*, 7, 5032, 2016.). And its training set contains the bond dissociation enthalpies of small organic molecules, these data have been calibrated by experimental measurements and high-level *ab initio* multireference methods (e.g. *J. Phys. Chem. A* 120, 4025, 2016.).

Fig. 1. Bond dissociation energy curves of the C-C and C-H bonds in ethane calculated by MN15 and MRCI+Q methods (8 active electrons and 8 active spaces.). The result shows that MN15 and MRCI+Q are in good agreement with each other.

3. While electronic excitation per se may not be that common, reactive recombination to form excited states is clearly a possibility. This should be mentioned. One example has been recently given for O+O recombination (Pezzella et al. (2020)) in which ground and excited state O₂ was formed upon recombination. As the reactivity of electronically excited O₂ can be several times larger towards carbo-hydrates, including electronically excited state in this example is clearly essential. What is the situation in the present work?

Reply: We agree that under certain reaction conditions, reactive recombination which can form excited states can be important. And we briefly discussed the work of Pezzella et al. in the revised manuscript. We carefully analyzed the trajectories and did not find O+O recombination in our simulation. Such a recombination reaction could occur at relatively low collision energies, but not that common at high temperatures such as in combustion.

4. Again, in their reply the authors point out the limitations of current electronic structure methods for treating excited states of large numbers of systems. This should be mentioned more explicitly in the manuscript. There are actually recent reactive MD studies of

nonadiabatic processes involving several electronic states, see Koner et al., Varandas et al., Schinke et al. and others, all on high-quality PESs.

Reply: Thank you for the suggestion. The limitations of current electronic structure methods for treating excited states are now mentioned more explicitly in the revised manuscript. And the recent works of Koner et al., Varandas et al, and Schinke et al. are mentioned and cited in the revised manuscript.

5. Point 2 from reviewer 1 is only incompletely answered. How different are the reaction conditions for the reactions generating OH from those used in the present simulations?

Reply: Firstly, experimental reaction conditions generating OH are performed at about 2250 K while the current simulation work is performed at 3000 K. Secondly, the rate constants in the GRI-Mech 3.0 database are not obtained from a single experimental measurement. They are actually obtained by combining and optimizing data from multiple experimental and theoretical works. For example, the rate constants of the $\cdot\text{CH}_3 + \text{H}_2\text{O} \rightarrow \text{CH}_4 + \cdot\text{OH}$ reaction was obtained by Cohen et al. (*Int. J. Chem. Kin.* 23, 397.) using theoretical optimizations based on the experimental results of Madronich et al (*20th Symp. (Int.) Combust., 1984, 703.*). The rate constant of the $\cdot\text{O} + \text{CH}_4 \rightarrow \text{CH}_3 + \cdot\text{OH}$ reaction was optimized by Tsang et al. on the basis of multiple works (*J. Phys. Chem. Ref. Data* 15, 1087.). Since there are no available experimental rate constants directly measured at 3000 K, we just used these two reaction rates given by the GRI-Mech 3.0 database at 3000 K (consistent with the temperature of MD) for comparison.

6. The authors refer to Arrhenius forms for the rate coefficients. In hypersonics, often a modified Arrhenius with an explicitly T-dependent prefactor is used. As this manuscript deals with combustion, such modified forms might offer some advantage and should be mentioned and discussed.

Reply: In our study, we calculated the rate constants by using a statistical method (Ref. 50 and Ref. 51.) and did not use the Arrhenius form to evaluate the rates. The T-dependent pre-factor Arrhenius formular is used by the GRI_Mech database to extrapolate the rate coefficients based on experiments and theoretical calculations. We mentioned it in the revised manuscript.

7. The reply to point 5, reviewer 1 is rather vague. The authors should state what density was used instead of reporting "..increased [the] density.." and similar for all other system variables. A short discussion of the differences between experimental reaction conditions and those used in the simulations is clearly necessary.

Reply: Thank you for this suggestion, we added a discussion on the difference between experimental and simulation conditions in the revised manuscript.

8. The figure from their reply letter on the energy conservation should be included at least in the SI. A larger scale for the energy has to be given and the simulation time needs to be considerably extended. Drifts in the total energy can not be seen from a 10 ps time series. Would the simulation leading to Figure 1 be suitable for that?

Reply: We extended the NVE simulation to 100 ps. As shown in the following figure, the total energy is conserved and the energy fluctuation is relatively small. We didn't extend to longer NVE simulation because it leads to unphysical high temperature as the simulation time increases due to the exothermic nature of the combustion as shown in the following figure. We also included this figure in the SI.

Fig. 2. Time dependences of the total energy (the relative value to the first snapshot) and temperature during the MD simulation under NVE ensemble.

9. While it is certainly true that finding new reactions is an attractive possibility of such a computational model, it should be stated explicitly that the accuracy of the model determines whether or not conclusions about how likely and feasible such new reactions are valid. To validate this, one such "new reaction" - e.g. the one forming cyclopropane - that was found must be investigated at a higher level of theory. From the reaction profile it should then be determined whether or not "finding this new reaction" is an artifact of the model because its accuracy is not appropriate or whether the "new reaction" is indeed feasible at the reaction conditions chosen in the simulations. Only then the conclusion from the reply letter "...demonstrates that reactive MD can be a powerful tool to study combustion reactions." is likely to be valid.

Reply: First, the current reactive MD based on the MN15 DFT functional has correctly reproduced almost all experimentally verified reactions (more than 100) in this combustion. The reactions that generate cyclopropene are not particularly different from other reactions already reproduced.

Based on the referee's suggestion, we also calculated the reaction energies of some typical reactions along the path toward the production of cyclopropene at the CCSD(T)/aug-ccpvtz level. These calculated CCSD(T) energies agree well with the MN15 energies, as shown in the following table. In view of the combustion reactions at high temperatures (3000K), the differences of a few kcal/mol in reactions energies are considered small and will not materially affect the results and conclusions.

Fig. 3. Reaction energies of seven typical reactions along the path to the production of *cyclopropene* at both the MN15 and CCSD(T) level.

10. In Figure 1 it is interesting to note that all concentrations are either monotonically increasing or decreasing. Is there an explanation for this?

Reply: These curves were smoothed to make them look better and clearer. In the revised manuscript, we mentioned this in the caption of Figure 1.

Minor point: the formatting of the references needs attention, e.g. Ref.33 has no page numbers, ditto for Ref. 16.

Reply: We are sorry for this mistake and these two references were corrected in the revised manuscript.

In summary, this work is a potentially useful way forward to investigate complex reactions in combustion. However, at several places the accuracy of the model needs to be more critically scrutinized, established, or demonstrated. Without critical validation it remains unclear whether or not the conclusions are sufficiently supported.

Reply: We believe that we have fully addressed the concerns about the accuracy of our computational model and any remaining doubt about the reliability of our results is cleared.

REVIEWERS' COMMENTS

Reviewer #1 (Remarks to the Author):

The authors have addressed all open points satisfactorily and the work is recommended for publication. This is a fine piece of work.